# Diagnostic Models Combining Clinical Information, Ultrasound and Biochemical Markers for Ovarian Cancer: Cochrane Systematic Review and Meta-Analysis

**DOI:** 10.3390/cancers14153621

**Published:** 2022-07-26

**Authors:** Clare F. Davenport, Nirmala Rai, Pawana Sharma, Jon Deeks, Sarah Berhane, Sue Mallett, Pratyusha Saha, Rita Solanki, Susan Bayliss, Kym Snell, Sudha Sundar

**Affiliations:** 1Test Evaluation Research Group, Institute of Applied Health Research, University of Birmingham, Birmingham B15 2TT, UK; p.sharma.3@bham.ac.uk (P.S.); j.deeks@bham.ac.uk (J.D.); s.berhane@bham.ac.uk (S.B.); susanbayliss26@gmail.com (S.B.); 2NIHR Birmingham Biomedical Research Centre, University Hospitals Birmingham NHS Foundation Trust, University of Birmingham, Birmingham B15 2TT, UK; 3Southend University Hospital NHS Trust, Southend-on-Sea SS0 0RY, UK; nimarai18@doctors.org.uk; 4Centre for Medical Imaging, University College London, London NW1 2BU, UK; sue.mallett@ucl.ac.uk; 5College of Medical and Dental Sciences, University of Birmingham, Birmingham B15 2TT, UK; pxs753@student.bham.ac.uk; 6Nuffield Division of Clinical Laboratory Sciences, John Radcliffe Hospital, Oxford OX3 9DU, UK; r.solanki@ndcls.ox.ac.uk; 7Centre for Prognosis Research, School of Primary, Community and Social Care, Keele University, Keele ST5 5BG, UK; k.snell@keele.ac.uk; 8Pan Birmingham Gynaecological Cancer Centre, City Hospital, Birmingham B187QH, UK; 9Institute of Cancer and Genomic Sciences, University of Birmingham, Vincent Drive, Edgbaston, Birmingham B152TT, UK

**Keywords:** ovarian cancer, RMI, ROMA, ADNEX, diagnostic test accuracy

## Abstract

**Simple Summary:**

Diagnosing ovarian cancer (OC) accurately helps triage patients to receive anticancer treatment and appropriate cancer surgery. We conducted a systematic review and meta-analysis evaluating models combining clinical information, biomarkers, and ultrasound to identify the most accurate test. Our review investigated 58 studies (30,121 patients, 9061 OC cases) and compared the standard of care test in the UK, Risk of Malignancy index I (RMI I) against risk of ovarian malignancy (ROMA) and Assessment of Different NEoplasias in the adnexa (ADNEX). Compared to RMI I, in pre-menopausal women, ROMA and ADNEX identified more cancers correctly (increased sensitivity) but increased the number of women classified as having cancer when they did not have cancer (reduced specificity). In post-menopausal women, ROMA identified more cancers than RMI I with similar specificity, whilst ADNEX identified the most cancers overall but with least specificity. Consideration should be given as to whether RMI I should be replaced by ROMA or ADNEX as the standard of care test for OC.

**Abstract:**

Background: Ovarian cancer (OC) is a diagnostic challenge, with the majority diagnosed at late stages. Existing systematic reviews of diagnostic models either use inappropriate meta-analytic methods or do not conduct statistical comparisons of models or stratify test performance by menopausal status. Methods: We searched CENTRAL, MEDLINE, EMBASE, CINAHL, CDSR, DARE, Health Technology Assessment Database and SCI Science Citation Index, trials registers, conference proceedings from 1991 to June 2019. Cochrane collaboration review methods included QUADAS-2 quality assessment and meta-analysis using hierarchical modelling. RMI, ROMA or ADNEX at any test positivity threshold were investigated. Histology or clinical follow-up was the reference standard. We excluded screening studies, studies restricted to pregnancy, recurrent or metastatic OC. 2 × 2 diagnostic tables were extracted separately for pre- and post-menopausal women. Results: We included 58 studies (30,121 patients, 9061 cases of ovarian cancer). Prevalence of OC ranged from 16 to 55% in studies. For premenopausal women, ROMA at a threshold of 13.1 (+/−2) and ADNEX at a threshold of 10% demonstrated significantly higher sensitivity compared to RMI I at 200 (*p* < 0.0001) 77.8 (72.5, 82.4), 94.9 (92.5, 96.6), and 57.1% (50.6 to 63.4) but lower specificity (*p* < 0.002), 92.5 (90.0, 94.4), 84.3 (81.3, 86.8), and 78.2 (75.8, 80.4). For postmenopausal women, ROMA at a threshold of 27.7 (+/−2) and AdNEX at a threshold of 10% demonstrated significantly higher sensitivity compared to RMI I at a threshold of 200 (*p* < 0.001) 90.4 (87.4, 92.7), 97.6 (96.2, 98.5), and 78.7 (74.3, 82.5), specificity of ROMA was comparable, whilst ADneX was lower, 85.5 (81.3, 88.9), 81.3 (76.9, 85.0) (*p* = 0.155), compared to RMI 55.2 (51.2, 59.1) (*p* < 0.001). Conclusions: In pre-menopausal women, ROMA and ADNEX offer significantly higher sensitivity but significantly decreased specificity. In post-menopausal women, ROMA demonstrates significantly higher sensitivity and comparable specificity to RMI I, ADNEX has the highest sensitivity of all models, but with significantly reduced specificity. RMI I has poor sensitivity compared to ROMA or ADNEX. Choice between ROMA and ADNEX as a replacement test will depend on cost effectiveness and resource implications.

## 1. Introduction 

This article is based on a Cochrane Review published in the Cochrane Database of Systematic Reviews (CDSR) 2022, Issue 7, DOI:10.1001/14651858. CD011964.pub2 *In press* (see www.cochranelibrary.com for information, accessed on 12 May 2022). [1] Cochrane Reviews are regularly updated as new evidence emerges and in response to feedback, and the CDSR should be consulted for the most recent version of the review.

Ovarian cancer (OC) has a high case fatality rate, largely attributed to the advanced stage at diagnosis in the majority of patients. Overall survival is 35% at 10 years; however, one year survival is only 51% at stage 4 compared to over 90% at stage 1 [2,3]. Globally, about 14–22% of women with OC will not receive anticancer therapy, probably due to late diagnosis impacting adversely on performance status and the ability to undergo treatment [4,5,6]. Diagnostic testing enables triage at two levels: initially from the community setting into hospital gynaecology clinics and then a further triage of some women to tertiary cancer care. Women classified as ‘low risk’ are managed by gynaecologists and can have fertility preservation, laparoscopic surgery, or surveillance whilst women classified as ‘high-risk’ are referred to a specialist cancer centre for surgery by expert gynaecological oncologists. Earlier, more accurate, diagnosis will enable better triage, optimisation of resources, and can improve outcomes. 

OC represents a diagnostic challenge: symptoms are non-specific [7] and prevalence in the community is low (the average primary care physician sees one patient with ovarian cancer in 400 patient encounters). In addition, OC comprises heterogenous histology types which undermines the potential accuracy of any single biomarker. Menopausal status is likely to be a significant modifier of test accuracy given the difference in pre-test probability and the differing spectrum of histological subtypes in pre- compared to post-menopausal women. Diagnosis of OC in premenopausal women is particularly challenging; the majority of tumours are benign and only 1 in 1000 symptomatic ovarian cysts is malignant. This increases to 3 in 1000 by age 50 [8]. False positive results are generated by ovarian physiology (cysts) and benign pathology, e.g., endometriosis. Given the above, a combination of tests is more likely to improve diagnostic accuracy over any single test alone. Current guidelines advocate the use of composite models which, in addition to adjustment for age or menopausal status, include multiple biomarkers. These include Risk of Malignancy Algorithm (ROMA) which combines biomarkers CA125 and He4, the Assessment of Different NEoplasias in the adneXa (ADNEX) model from the International Ovarian tumour analysis consortium (IOTA) and the Risk of Malignancy Index I (RMI I) which both combine Ca125 and ultrasound characteristics (ACOG guidelines). ADNEX also offers a polynomial probability of histology. RMI I is recommended by the National Institute for Health and Clinical Excellence [9] and the Royal College of Obstetrics and Gynaecology in the UK [10]. 

We conducted a systematic review to compare the accuracy of diagnostic models for the diagnosis of OC, distinguishing between pre- and post-menopausal women. Our target population was women presenting with symptoms or signs suspicious for ovarian cancer. We used test accuracy review methods recommended by the Cochrane collaboration. 

## 2. Methods and Materials

We followed Cochrane Screening and Diagnostic Tests Methods group recommendations [11].

Review registration number: Rai N, Champaneria R, Snell K, et al. Symptoms, ultrasound imaging and biochemical markers alone or in combination for the diagnosis of ovarian cancer in women with symptoms suspicious of ovarian cancer. Cochrane Database Syst Rev. 2015; 2015(12):CD011964. Published 7 December 2015. doi:10.1002/14651858.CD011964.

### 2.1. Criteria for Including Studies

We included studies in women aged 18 years or older undergoing testing in any healthcare setting (primary/secondary/tertiary) for a suspicion of OC. Studies investigated the accuracy of RMI I, ROMA, or ADNEX at any test positivity threshold. We included comparative and non-comparative diagnostic test accuracy studies if they verified index test results against histology or clinical follow-up and with sufficient data to extract 2 × 2 diagnostic tables separately for pre- and post-menopausal women. We included studies of any design, including case-control studies, where controls included benign ovarian pathology.

We excluded studies restricted to pregnant women, history of OC, recurrent, metastatic OC, and screening studies. 

### 2.2. Search Strategy

We used sensitive search strategies combining terms for the target condition (OC), biomarkers, symptom indices and ultrasound, and terms to describe diagnostic models and algorithms. We searched Cochrane Central Register of Controlled Trials (CENTRAL), MEDLINE and MEDLINE In Process (Ovid), EMBASE (Ovid), CINAHL (Ebsco), the Cochrane Database of Systematic Reviews (CDSR), Database of Abstracts of Reviews of Effects (DARE), Health Technology Assessment Database (HTA), and SCI Science Citation Index (ISI Web of Knowledge) with a date restriction (1991 onwards) to ensure applicability to current technology. Trials registers, conference abstracts, and proceedings were searched for unpublished studies without date restrictions. Reference lists of systematic reviews and guidelines were searched. No language restrictions were applied. Searches were completed in June 2019 (Appendix A). 

### 2.3. Data Extraction and Quality Assessment

Selection, data extraction, and quality assessment were carried out independently and in duplicate by 2 reviewers, with disagreements resolved by a third reviewer. Quality assessment was through a QUADAS-2 checklist tailored to the topic [12]. Tailoring included a question about follow up of index test negatives (>12 months), two questions relevant to multivariable model/composite index test validation studies from a quality checklist for prognostic studies [13], and whether borderline ovarian tumours (BOT) were appropriately handled, i.e., BOT categorised as malignant/index test positive for analysis. Key elements of applicability included symptoms as reason for testing, inclusion of comorbidity, e.g., endometriosis, that might affect estimates of test accuracy, and expertise of clinicians performing ultrasound. The modified QUADAS-2 tool used is available from the authors on request. 

### 2.4. Statistical Analysis

Exploratory analyses included plotting estimates of sensitivity and specificity separately in pre- and post-menopausal women and grouped by test threshold on Forest plots and for summary ROC plots. Analyses were conducted in Stata version SE 17.0 [14] and SAS software (version 9.4) [15]. HSROC models were fitted using the NLMIXED procedure in SAS. Bivariate models were fitted using the *meqrlogit* command in Stata. In cases where both random effects were set to zero, a fixed effect logistic regression was fitted using the *blogit* command. Absolute differences in sensitivities/specificities and *p*-values were derived from bivariate models using the *nlcom* command in Stata. This computes point estimates and standard errors using the delta method.

### 2.5. Estimation of Accuracy for Individual Tests

Where adequate data were available and considered reasonable to pool results, we performed meta-analyses using hierarchical models including random effects [16,17]. To estimate average sensitivity and specificity at fixed thresholds, we performed analysis of each index test version by first restricting to studies that reported thresholds recommended in guidelines and/or used in clinical practice and secondly to commonly reported thresholds. For ROMA, we included studies using thresholds +/−2 units around most commonly reported thresholds. We used random-effects univariate analyses where pooling was considered an appropriate approach, but bivariate models failed to converge. Where appropriate, models were simplified by setting near-zero variance estimates of the random effects to zero [18].

### 2.6. Comparison of Test Accuracy

Since studies reported different thresholds per index test to define disease positivity, hierarchical summary ROC (HSROC) models [19] that included random-effects parameters for variation in accuracy and threshold were fitted to maximise use of all date available. Comparisons between index tests were made by adding a covariate for test type to the accuracy and threshold parameters, while assuming a common underlying shape. A summary ROC curve for each index test across all included thresholds was estimated. Each included study contributed one threshold to the summary ROC curve. Where an individual study reported more than one threshold, the most commonly reported threshold for that index test across all included studies was selected for the meta-analyses. P values for difference in accuracy for each test compared to RMI I (RMI I being recommended test for routine use in the United Kingdom) were computed using Wald tests. The difference in sensitivities at fixed specificities of 80% and 90% for each index test compared to RMI I with 95% CI was also reported. 

To illustrate comparative accuracy at specific test operating thresholds, we also undertook a comparison of index tests at the single most commonly reported threshold using bivariate hierarchical models that included a covariate for test type. Absolute differences in sensitivity/specificity and the corresponding *p*-values for each pairwise test comparison were reported from the model. Where appropriate, models were simplified by assuming no correlation between sensitivity and specificity estimates or by setting near-zero variance estimates of the random effects to zero [18].

We translated summary estimates of sensitivity and specificity into summary estimates of the absolute numbers of true positives, false negatives, false positives, and true negatives. We did this using a hypothetical population of 1000 women using an estimate of disease prevalence (pre-test probability) reflecting the NICE threshold for cancer referral from primary care to hospital settings in the UK of 3% (Appendix A) (NICE 2017) [20].

### 2.7. Investigation of Heterogeneity

We investigated the effect on test accuracy of menopausal status (pre-menopausal or post-menopausal) and classification of BOT as disease positive (grouped with malignant ovarian tumours) or where classification of BOT was unclear or BOT were excluded. Grouping of BOT with OC was considered appropriate (reflecting current surgical practice) whereas exclusion of BOT was considered methodologically inappropriate. Estimation of differences in accuracy were performed using the NLMIXED procedure in Statistical Analysis System [15] by including borderline grouping as covariates in the bivariate model. Differences in accuracy were reported using the ratio of diagnostic odds ratios with 95% CI and associated *p* values using Wald tests.

## 3. Results

### 3.1. Quantity and Quality of Evidence

#### 3.1.1. Quantity of Evidence

A total of 52,099 unique records were identified. After reviewing titles and abstracts, full-text screening revealed 1215 potentially relevant studies of which 58 studies reporting 66 datasets were eligible for inclusion. The most common reason for exclusion was lack of data to populate a 2 × 2 test accuracy table (*n* = 260) (Figure 1).

In total, 49 studies investigated a single test whilst only 10 included a within person comparison of 2 or more index tests [21,22,23,24,25,26,27,28]. Test types and thresholds were too varied to permit separate meta-analyses of direct comparison studies. Included studies report on the accuracy of RMI I at thresholds of 200 (*n* = 17) and 250 (*n* = 2), 10,283 participants, 2654 OC), ROMA (42 studies, 13,715 participants, 3944 OC) at threshold pairs for pre- and post-menopausal women of 7.4 (+/−2) (*n* = 12) and 25.3 (+/−2) (*n* = 15); 12.5 and 14.4 (*n* = 3), 13.1 (+/−2) (*n* = 27), and 27.7 (+/−2) (*n* = 13); 11.4 (*n* = 11) and 29.9 (*n* = 12) and ADNEX (4 studies, 3061 participants, 1204 cases of ovarian cancer) to achieve a post-test probability of ovarian cancer of either 3%, 5%, 10%, and 15%.

#### 3.1.2. Characteristics of Included Studies

Appendix A presents characteristics of included studies. 

Studies were conducted in Europe (40), Asia-Pacific (12), North America (5), and South America (1). Eighteen were multicentre. Forty-eight studies were conducted in a hospital (8 mixed secondary and tertiary, 28 tertiary, and 12 secondary) and in 10 healthcare setting was not reported. Clinical pathway to test was not described in any studies. Prevalence of OC ranged between 16% (RMI I, ROMA), and 27% (AdNEX) in pre-menopausal and between 38% (RMI I), 45% (ROMA) and 55% (AdNEX) in post-menopausal women. Four studies specified presence of symptoms whilst 10 ROMA studies reported that an adnexal mass was identified following imaging. Eighteen ROMA and four RMI I studies explicitly restricted inclusion to epithelial ovarian cancer (EOC) and seven ROMA and one RMI I study explicitly excluded BOT. In a further 18 ROMA and 3 RMI I studies occurrence of BOT was not reported. The majority of studies were conducted in specialist gynaecological oncology centres (36/58) in women scheduled for surgery. 

#### 3.1.3. Quality (Risk of Bias and Applicability Concerns of Included Studies)

Risk of bias:

Figure 2 illustrates quality assessment for 58 included studies (reporting 66 datasets) across participant, index tests, reference standard, and flow and timing domains. For participant selection domain, 15/58 (27%) studies were at high risk of bias and 38/59 (64%) at unclear risk of bias. Only 5 studies were judged at low risk of bias [24,25,26,27,28] where authors explicitly reported consecutive sampling and tumour pathology such that impact on accuracy could be investigated. For index test domain, the majority of ROMA (33/42; 79%), (2/4) ADNEX and 9/20 (45%) of RMI I studies were judged low risk of bias as they were prospective or because of the objective nature of the index test (ROMA). One RMI I study [29] was judged at high risk of bias because RMI I results were interpreted with knowledge of reference standard. Four ROMA studies [30,31,32,33] were judged at high risk of bias because no threshold was pre-defined. For reference standard domain, the majority of studies 30/58 (52%) were judged low risk of bias. In total, 27 of 58 studies (47%) were judged unclear and two at high risk of bias [34,35] either because minimum length of follow up for index negatives was not reported or because there was concern that the reference standard was ascertained with knowledge of the index test result. For flow and timing domain, the majority of studies were judged unclear risk of bias 32/58 (55%), most commonly because of no information regarding the interval between index test and reference standard. In total, 12 of 58 (21%) studies were judged at high risk of bias because not all participants receiving an index test received a reference standard. 

#### 3.1.4. Applicability

In total, 53 of 58 studies (91%) were judged high or unclear applicability for participant selection domain because participants did not explicitly represent symptomatic women. For index test domain, applicability was high or unclear for all RMI and ADNEX studies because ultrasound was conducted by specialist sonographers or level of specialism was unclear. For reference standard domain, applicability was judged as high or unclear in 49/58 (85%) studies because BOT had been excluded from analysis or classification of BOT was unclear. 

### 3.2. Estimation of Test Accuracy

A consistent difference was observed in sensitivity (higher in post-menopausal women) and specificity (lower in post-menopausal women) across index tests operating at a fixed threshold (Appendix A) and across all versions of all index tests at all thresholds analysed. Subsequently, sensitivity and specificity were estimated in pre- and post-menopausal women separately. 

#### 3.2.1. Estimation of Accuracy—Single Tests

Appendix A summarises accuracy of 58 unique included studies (66 data sets) in pre- and post-menopausal women. ROMA and ADNEX studies included reported multiple test positivity thresholds. For ROMA, no evidence of a significant difference in accuracy at thresholds was identified and we chose the threshold reported by most studies (13.1 (+/−2) and 27.7 (+/−2) in pre- and post-menopausal women, respectively). For ADNEX, the only common threshold used across all 4 included studies was that to achieve a post-test probability of OC of 10%.

#### 3.2.2. Comparison of Test Accuracy: HSROC Analysis

Figure 3A illustrates comparison of test performance contributed by all studies at all test positivity thresholds in pre-menopausal women. In pre-menopausal women, ADNEX (*p* = 0.0083) but not ROMA (*p* = 0.5608) demonstrates superior accuracy compared to RMI I. Figure 3B illustrates that in post-menopausal women ROMA (*p* = 0.0043) but not ADNEX (*p* = 0.0522) demonstrates superior overall accuracy compared to RMI I.

We also compared the sensitivity of tests when fixing specificity at 80% and 90% (Table 1) in keeping with clinical consensus [8,10]. Of note the sensitivity estimate for ADNEX in pre-menopausal women at a fixed specificity of 80% and in pre- and post-menopuasal women at a fixed specificity of 90% is extrapolating beyond the data from included ADNEX studies (Figure 3B). 

In pre-menopausal women at a fixed specificity of 80%, RMI I has sensitivity of 79.8 (69.7 to 87.1). The average difference in sensitivity of ROMA compared to RMI I at a fixed specificity of 80% is compatible with chance (2.4 (−5.8 to 10.5) but a significant increase in average sensitivity is observed with ADNEX: 15.1 (5.7 to 24.5) (Table 1).

In post-menopausal women at a fixed specificity of 80%, RMI I has a sensitivity of 85.4 (81.1 to 88.9). ROMA and ADNEX both demonstrated a significant increase in average sensitivity compared to RMI of 5.5 (1.7 to 9.4) and 8.8 (2.4 to 15.1), respectively. (Table 1).

#### 3.2.3. Comparison of Test Accuracy at Fixed Test Positivity Thresholds

Table 2 and Table 3 demonstrates test performance contributed by the majority of studies for each index test: ROMA at a threshold of 13.1 (+/−2), (27/42 studies) and 27.7 (+/−2) 13/42 studies) in pre- and post-menopausal women; ADNEX at a post-test probability of 10% (4/4 studies in pre- and post-menopausal women) compared to RMI I at a threshold of 200 (17/19 studies in pre- and post-menopausal women). 


*Pre-menopausal women: Test performance of ROMA and ADNEX compared to RMI I specific thresholds.*


Table 2: In pre-menopausal women, RMI I at 200 (17 studies, 5233 participants, 851 cases of OC) had a sensitivity of 57.2% (49.9 to 64.2) and specificity of 92.5 (90.4 to 94.2). Compared to RMI I: ROMA at a threshold of 13.1 (+/−2) (27 studies, 4463 participants, 825 cases OC), demonstrated a significant increase in sensitivity 20.3 (11.8 to 28.9) but a significant decrease in specificity −8.2% (−11.7 to −4.8) and ADNEX at 10% (4 studies, 1696 participants, 455 cases OC), demonstrated a significant increase in sensitivity 38.3% (30.5 to 46.3 but a significant decrease in specificity −14.8% (−23.9 to −5.6).

Based on this analysis, in a clinical setting with a pre-test probability of ovarian cancer of 3%, for every 1000 pre-menopausal women tested (see Appendix A):Consequences of a positive test result:-An estimated 90 will have an RMI I result indicating ovarian cancer is present and of these 73 (81%) will not have ovarian cancer. -An estimated 176 will have a ROMA result indicating ovarian cancer is present and of these 152 (86%) will not have ovarian cancer. -An estimated 245 will have an ADNEX result indicating ovarian cancer is present and of these 216 (88%) will not have ovarian cancer. Consequences of a negative test result:-Of the 910 people with an RMI I result indicating that ovarian cancer is not present, 13 (1%) will actually have ovarian cancer.-Of the 824 people with a ROMA result indicating that ovarian cancer is not present, 7 (0.8%) will actually have ovarian cancer.-Of the 755 people with an ADNEX result indicating that ovarian cancer is not present, 1 (0.1%) will actually have ovarian cancer.



*Post-menopausal women: Test performance or ROMA and ADNEX compared to RMI I at specific thresholds.*


Table 3: In post-menopausal women, RMI I (17 studies, 4369 participants, 1664 cases of OC) had sensitivity of 78.5% (74.6 to 82.0) and a specificity of 85.4 (82.1 to 88.3). Compared to RMI I: ROMA at a threshold of 27.7 (+/−2) (13 studies, 2002 participants, 852 cases OC), demonstrated a significantly increased sensitivity 11.9% (7.6 to 16.4) but comparable specificity −4.0% (−9.4 to 1.5), ADNEX at threshold to achieve post-test probability of OC of 10% (4 studies, 1365 participants, 749 cases OC), demonstrated significantly increased in sensitivity 19.1% (15.1 to 23.1) but a significant decrease in specificity −30.5% (−43.0 to −17.9).

Based on this analysis, in a clinical setting with a pre-test probability of ovarian cancer of 3%, for every 1000 post-menopausal women tested (see Appendix A):Consequences of a positive test result:-An estimated 165 will have an RMI I result indicating ovarian cancer is present and of these 142 (86%) will not have ovarian cancer. -An estimated 207 will have a ROMA result indicating ovarian cancer is present and of these 179 (86%) will not have ovarian cancer. -An estimated 466 will have an ADNEX result indicating ovarian cancer and of these 437 (94%) will not have ovarian cancer. Consequences of a negative test result:-Of the 835 people with an RMI I result indicating that ovarian cancer is not present, 6 (0.7%) will actually have ovarian cancer. -Of the 793 people with a ROMA result indicating that ovarian cancer is not present, 3 (0.4%) will actually have ovarian cancer. -Of the 534 people with an ADNEX result indicating that ovarian cancer is not present, 1 (0.1%) will actually have ovarian cancer. 

### 3.3. Investigation of Heterogeneity: The Effect of Classification of BOT on Estimates of Test Accuracy

In pre-menopausal women (ROMA *n* = 38 studies; RMI I 19 studies) and post-menopausal women (ROMA *n* = 40 studies) there was sufficient data, when utilising all test positivity thresholds at a fixed specificity of 80%, to allow comparison of sensitivity estimated by studies where BOT were classified as positive (grouped with malignant tumours) against studies excluding BOT from analysis or where classification of BOT was unclear. In post-menopausal women, for ROMA, a significant decrease in sensitivity of 6.4% (1.2 to 11.5) was observed for studies grouping BOT with malignant compared to studies where BOT were excluded or where categorisation for analysis was unclear (Appendix A). 

## 4. Discussion

This systematic review highlights that the performance of test combinations varies considerably between pre- and post-menopausal women. Evaluation of models should include an assessment of their performance separately in these different target populations, including when models include variables to adjust for this characteristic. In pre-menopausal women, both ROMA and ADNEX demonstrated significantly higher sensitivity than RMI, ADNEX demonstrating marginally significantly higher sensitivity compared to ROMA. Both ROMA and ADNEX showed significantly lower specificity than RMI. In post-menopausal women, both ROMA and ADNEX demonstrated significantly higher sensitivity than RMI I. ROMA demonstrated comparable specificity to RMI I, whilst for ADNEX specificity was significantly lower compared to RMI I. 

### 4.1. Strengths 

Our systematic review addresses several issues in existing systematic reviews. As part of a scoping review, 10 original systematic reviews were identified up to 2021, [9,36,37,38,39,40,41,42,43,44] NICE 2011; 6 of 11 reviews included ROMA, 7 RMI I. The most recent review search date was 2018 [42]. None included ADNEX. Two compared ROMA and RMI I [42,44], 4 compared RMI I and LR2 [36,38,40,44], whilst 6 reviewed single tests. In total, 4 of 11 reviews did not present results separately for pre- and post-menopausal [40] women. In total, 9 of 11 reviews undertook meta-analysis but only 5 used appropriate statistical methods (hierarchical modelling) [11]. 

Novel features of this review include systematic investigation of effects of menopausal status, investigation of BOT on estimates of test accuracy and statistical comparison of test accuracy between models. We mitigated heterogeneity by restricting analysis to primary adnexal tumours and where this was not possible or unclear in studies with mixed primary, recurrent, and metastatic disease, this was reflected in downgrading of quality.

### 4.2. Limitations 

We acknowledge limitations of the search date, in recognition of the relatively small number of ADNEX studies included we performed a scoping search for primary studies published since our search cut-off date of June 2019. Three studies were found, two from single centres [45,46] and one multicentre study [47].

Only one study [45] reported sensitivity and specificity separately in pre- and post-menopausal women. Sensitivity and specificity were both 83% in premenopausal women and 100% and 76%, respectively, in postmenopausal women at a threshold to achieve a post-test probability of ovarian cancer of 10%. These estimates are in line with studies included in this review. Inclusion of this single eligible additional ADNEX study is unlikely to alter the conclusions of this review.

Deficiencies of included studies included lack of data and poor reporting precluding quality assessment and investigation of potential important sources of heterogeneity. These included clinical setting, target condition, histological subtype and stage; included studies included a varying range of ovarian pathology. A lack of distinction between pre- and post-menopausal women when evaluating test accuracy is a major limitation of research in this area.

### 4.3. Recommendations for Clinical Practice

Guidelines that recommend RMI I for diagnosis of OC in pre- and post-menopausal women should be changed. Whether ROMA or ADNEX should replace RMI I will depend on how health care systems view the trade-off between sensitivity and specificity. Implementing ultrasound models will require training in specialist ultrasound skills and quality assurance processes, similar to those introduced for nuchal scans in early pregnancy. Implementing ultrasound testing through dedicated ‘pelvic mass’ clinics may represent a method for achieving this and this is currently being investigated within the ROCkeTS study [48]. Implementing testing with ROMA will require investment in laboratory processes.

### 4.4. Recommendations for Research

Most included studies demonstrated a high prevalence of OC (16–55%) in both pre- and post-menopausal women reflective of tertiary hospitals and a highly preselected population. Research is urgently needed to evaluate tests for diagnosis of ovarian cancer in community settings and lower prevalence. The ROCkeTS study will report on a head-to-head comparison of diagnostic tests in a lower prevalence setting than previously published [48]. Future systematic reviews should present results stratified by menopausal status.

## 5. Conclusions

Based on the results of this review RMI I should not be recommended for the diagnosis of OC in pre- or post-menopausal women. The test of choice between ROMA and ADNEX will depend on health care systems’ evaluation of the trade-off between sensitivity and specificity.

## Figures and Tables

**Figure 1 cancers-14-03621-f001:**
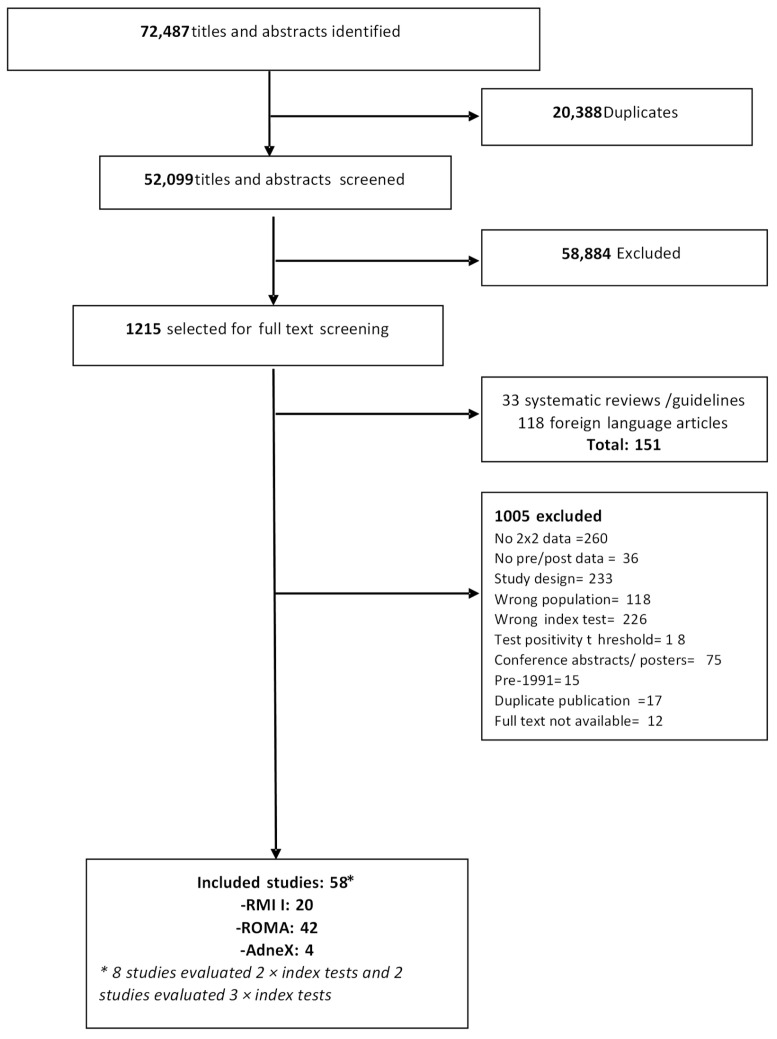
PRISMA diagram of included studies.

**Figure 2 cancers-14-03621-f002:**
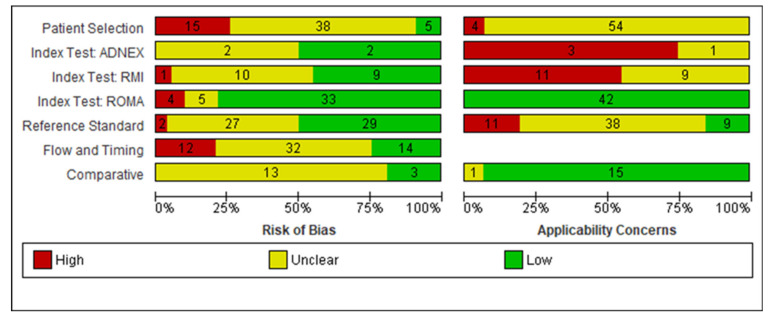
Quality assessment (QUADAS-2) of included studies.

**Figure 3 cancers-14-03621-f003:**
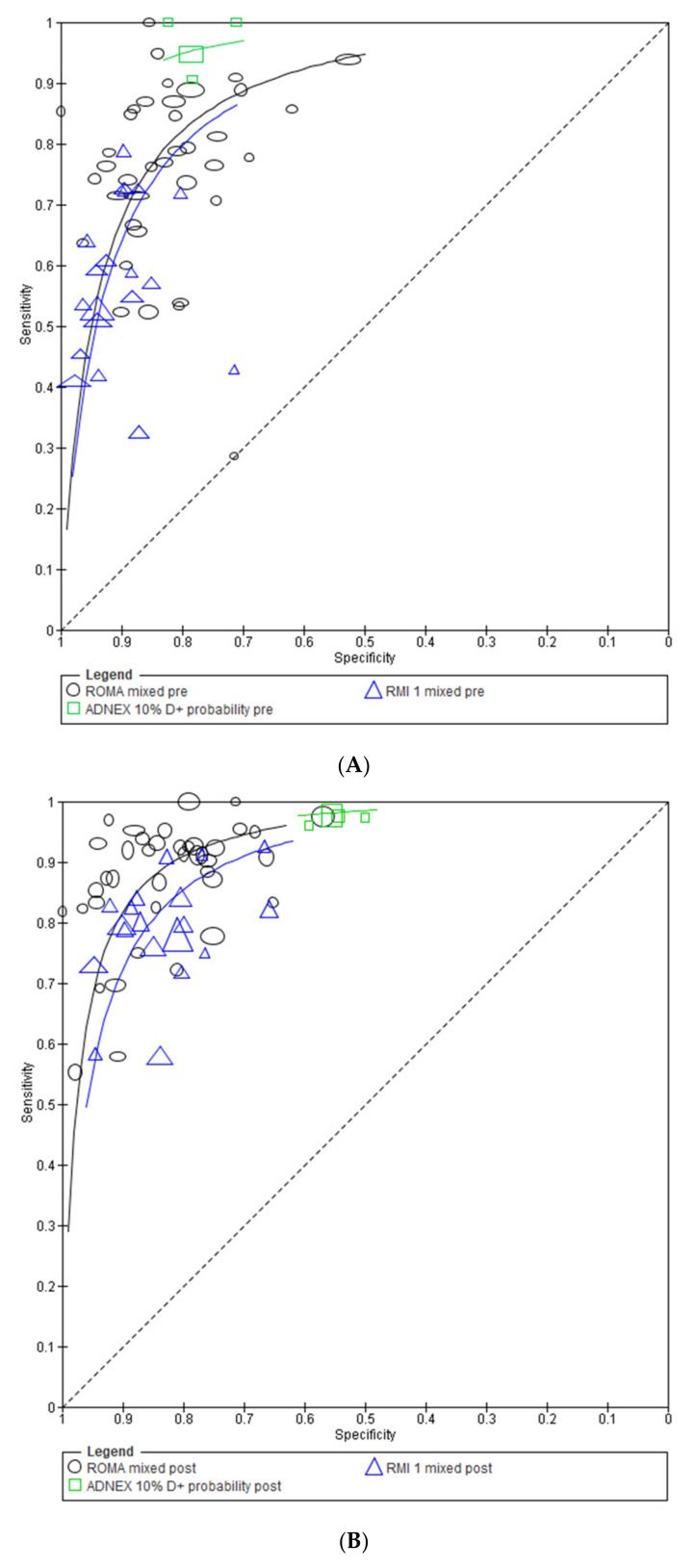
(**A**) HSROC plot of the performance of RMI I, ROMA, and ADNEX (all reported thresholds, all included studies) in pre-menopausal women. (**B**) HSROC plot of the performance of RMI I, ROMA, and ADNEX (all reported thresholds, all included studies) in post-menopausal women.

**Table 1 cancers-14-03621-t001:** HSROC analysis: comparison of ROMA and ADNEX compared to RMI I. Mixed test positivity threshold analysis at fixed specificities of 80% and 90%.

Test	Studies	Participants	OC Cases	Diagnostic Odds Ratio (95% CI)	Relative Diagnostic Odds Ratio (95% CI)	*p*-Value	Sensitivity at Fixed Specificity of 80%		Sensitivity at Fixed Specificity of 90%	
							Sensitivity (95% CI)	Difference from RMI 1 (95% CI)	Sensitivity (95% CI)	Difference from RMI 1 (95% CI)
**Pre-menopausal**										
RMI 1 200/250	19	5694	893	15.8 (9.2, 27.1)	-		79.8 (69.7, 87.1)	-	64.0 (55.9, 71.4)	-
ROMA mixed	38	7616	1198	18.4 (14.2, 23.8)	1.17 (0.69, 1.99)	0.5608	82.1 (77.9, 85.7)	2.4 (−5.8, 10.5)	67.5 (60.0, 74.2)	3.5 (−8.0, 14.9)
ADNEX 10%	4	1696	455	74.8 (31.2, 179.4)	4.75 (1.52, 14.81)	0.0083	94.8 (89.1, 97.7)	15.1 (5.7, 24.5)	89.3 (77.7, 95.2)	25.2 (13.3, 37.1)
**Post-menopausal**										
RMI 1 200/250	19	4589	1761	23.5 (17.6, 31.3)	-		85.4 (81.1, 88.9)	-	72.3 (65.8, 78.0)	-
ROMA mixed	40	6099	2746	40.3 (31.6, 51.3)	1.72 (1.19, 2.47)	0.0043	91.0 (88.8, 92.7)	5.5 (1.7, 9.4)	81.8 (76.8, 85.8)	9.5 (3.5, 15.4)
ADNEX 10%	4	1365	749	64.9 (24.7, 170.4)	2.77 (0.99, 7.73)	0.0522	94.2 (87.3, 97.4)	8.8 (2.4, 15.1)	87.8 (72.9, 95.1)	15.5 (5.2, 25.9)

Notes to table: ADNEX 10%: threshold to achieve a post-test probability of ovarian cancer of 10%. ADNEX studies reported a range of thresholds but all included a threshold of 10%. For RMI I and ROMA studies, each included study contributed a different test positivity threshold.

**Table 2 cancers-14-03621-t002:** Bivariate model-pairwise comparisons: pre-menopausal women.

Absolute Sensitivity Difference (95% CI), *p*-Value for Comparison Absolute Specificity Difference (95% CI), *p*-Value for Comparison			RMI I (200)	ROMA (13.1 ± 2)
		**Studies (Participants)**	17 (4369)	13 (2002)
		**Sensitivity % (95% CI)** **Specificity % (95% CI)**	78.5 (74.6, 82.0) 85.4 (82.1, 88.3)	90.4 (87.4, 92.7) 81.5 (76.5, 85.6)
	**Studies (participants)**			
**ROMA** (27.7 ± 2)	13 (2002)	90.4 (87.4, 92.7) 81.5 (76.5, 85.6)	11.9 (7.3, 16.4), *p* < 0.0001 −4.0 (−9.4, 1.5), *p* = 0.155	-
**ADNEX** (10)	4 (1365)	97.6 (95.4, 98.8) 55.0 (42.8, 66.6)	19.1 (15.1, 23.1), *p* < 0.0001 −30.5 (−43.0, −17.9), *p* < 0.0001	7.2 (4.2, 10.3), *p* < 0.0001 −26.5 (−39.5, −13.6), *p* < 0.0001

**Table 3 cancers-14-03621-t003:** Bivariate model-pairwise comparisons: pre-menopausal women.

Absolute Sensitivity Difference (95% CI), p-Value for Comparison Absolute Specificity Difference (95% CI), p-Value for Comparison			RMI I (200)	ROMA (13.1 ± 2)
		**Studies (Participants)**	17 (5233)	27 (4463)
		**Sensitivity % (95% CI)** **Specificity % (95% CI)**	57.2 (49.9, 64.2) 92.5 (90.4, 94.2)	77.6 (72.6, 81.8) 84.3 (81.2, 86.9)
	**Studies (Participants)**			
**ROMA** (13.1 ± 2)	27 (4463)	77.6 (72.6, 81.8) 84.3 (81.2, 86.9)	20.3 (11.8, 28.9), *p* < 0.0001 −8.2 (−11.7, −4.8), *p* < 0.0001	-
**ADNEX** (10)	4 (1696)	95.6 (90.8, 97.9) 77.7 (67.6, 85.4)	38.4 (30.5, 46.3), *p* < 0.0001 −14.8 (−23.9, −5.6), *p* = 0.002	18.0 (12.4, 23.6), *p* < 0.0001 −6.5 (−15.9, 2.9), *p* = 0.172

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
