# Peer review of "Diagnostic Models Combining Clinical Information, Ultrasound and Biochemical Markers for Ovarian Cancer: Cochrane Systematic Review and Meta-Analysis"

_cancers, 2022, doi:10.3390/cancers14153621_

Round 1

Reviewer 1 Report

The authors of "A systematic review and meta-analysis of diagnostic models combining clinical information, ultrasound and biochemical markers for ovarian cancer" compiled a very thorough analysis of predictive models to determine OC in women at various stages of their lives (i.e. pre and post menopausal). Following the combination and meta-data analysis, a comparative analysis of these methods was carried out, which is of extreme interest to the medical and scientific communities. Below are a few considerations for improvement that were detected during review.

1) Several times throughout the manuscript the authors provide extensive numbers and analytics in text, which can be very overwhelming and cumbersome to sort through (such as in the abstract). Can these instances be limited to include predominantly in tables and graphs? Also including the relevance of these statistics would aid in comprehension and reviewer interest.

2) Due to the limited number of included ADNEX trials, this should be excluded.

3) The tables are overwhelming and should be reformatted for consistency and clarity. 

Reviewer 2 Report

Critical evaluation and re-evaluation of diagnostic measures in oncology is important for further improvement of diagnostic approaches.  High mortality rate among ovarian cancer patients is directly related to the low efficiency of diagnosis.

 Good systematic review with well-described inclusion criteria for studies and a large body of analytical work. A few grammatical errors need to be corrected and attention paid to:

 1. Conflicting data:

-Table 1 «*8 studies evaluated 2 x index tests and 2 studies evaluated 3 x index tests»- 10 tests in total;

- Line 203 «Forty nine studies investigated a single test whilst only 9 included a within person 203 comparison of 2 or more index tests».

 2. Linу 286 – wrong link to Figure 3?

Reviewer 3 Report

Thank you for submitting the article to the journal.

The article is very comprehensive and easy to read. It very nicely reviews all the relevant information necessary in clinical practice.

I have a few suggestions which would further enhance the article.

1.     The discussion section is suboptimal, with only strengths, limitations, and recommendations. It needs to be more elaborative. May I suggest some points, but please be free to add yours too—any similar previous studies in the literature, and if their study results are concordant with yours. Also, you can describe some relevant studies in the analysis and if your results are concordant or discordant with their results. 

2.     The article uses heterogenicity. Can you also add any publication bias analysis if performed?

3.     The title and analysis mention ultrasound. Ovarian cancer is hard to diagnose on ultrasound. They can be screening tools but never diagnostic. There are two ways around this: either modify the term as “screening ultrasound” or, if feasible, analyze on more diagnostic imaging modalities such as CT or MRI. There are many metanalyses in the literature on CT and MRI.  
